# Inversion Method for Transformer Winding Hot Spot Temperature Based on Gated Recurrent Unit and Self-Attention and Temperature Lag

**DOI:** 10.3390/s24144734

**Published:** 2024-07-21

**Authors:** Yuefeng Hao, Zhanlong Zhang, Xueli Liu, Yu Yang, Jun Liu

**Affiliations:** 1School of Electrical Engineering, Chongqing University, Chongqing 400044, China; 20201101076g@stu.cqu.edu.cn (Y.H.); 20153560@cqu.edu.cn (X.L.); 20163724@cqu.edu.cn (Y.Y.); 2Chengdu Power Supply Company, State Grid Sichuan Electric Power Company, Chengdu 610041, China; 3Electric Power Science Research Institute, Guizhou Power Grid Co., Ltd., Guiyang 550002, China; liujj1@hnu.edu.cn

**Keywords:** winding hotspot temperature, temperature lag, mutual information (MI), SA-GRU, inversion method

## Abstract

The hot spot temperature of transformer windings is an important indicator for measuring insulation performance, and its accurate inversion is crucial to ensure the timely and accurate fault prediction of transformers. However, existing studies mostly directly input obtained experimental or operational data into networks to construct data-driven models, without considering the lag between temperatures, which may lead to the insufficient accuracy of the inversion model. In this paper, a method for inverting the hot spot temperature of transformer windings based on the SA-GRU model is proposed. Firstly, temperature rise experiments are designed to collect the temperatures of the entire side and top of the transformer tank, top oil temperature, ambient temperature, the cooling inlet and outlet temperatures, and winding hot spot temperature. Secondly, experimental data are integrated, considering the lag of the data, to obtain candidate input feature parameters. Then, a feature selection algorithm based on mutual information (MI) is used to analyze the correlation of the data and construct the optimal feature subset to ensure the maximum information gain. Finally, Self-Attention (SA) is applied to optimize the Gate Recurrent Unit (GRU) network, establishing the GRU-SA model to perceive the potential patterns between output feature parameters and input feature parameters, achieving the precise inversion of the hot spot temperature of the transformer windings. The experimental results show that considering the lag of the data can more accurately invert the hot spot temperature of the windings. The inversion method proposed in this paper can reduce redundant input features, lower the complexity of the model, accurately invert the changing trend of the hot spot temperature, and achieve higher inversion accuracy than other classical models, thereby obtaining better inversion results.

## 1. Introduction

Electricity plays an important role in the national economy [1,2,3], and regarding transformers, as the core components of power transmission and distribution, their safety and stability are crucial for ensuring the normal operation of the power system. The thermal condition of power transformers is a vital indicator for assessing insulation performance. If the winding hotspot temperature is too high, it will affect the equipment’s voltage withstand capability and mechanical strength, leading to breakdown accidents [4,5]. Therefore, the inversion of transformer winding hotspot temperature is crucial for predicting equipment operating conditions effectively and enabling predictive maintenance to achieve precise operation and maintenance.

The existing research methods for obtaining transformer hot spot temperature mainly include the direct measurement method, thermal circuit model method, numerical simulation method, empirical formula method, etc. However, these methods have some limitations in use due to reliability, efficiency, accuracy, and other issues.

Due to its remarkable ability to extract features from various complex data, machine learning has been widely applied in the study of transformer winding hotspot temperature. To date, many scholars have conducted relevant research on the prediction of winding hotspot temperature [6,7,8,9]. Deng et al. [10] predicted the hotspot temperature of a 10 kV transformer using support vector regression (SVR) based on primary feature temperature points obtained from thermal field calculations on the transformer casing. Sun et al. [11] combined actual operating temperature, load, the transformer cooling method, and ambient temperature data from monitored 35 kV dry-type transformers, optimizing SVR models using particle filtering for predicting transformer winding hotspot temperature. Wei et al. [12] considered factors such as sunlight and external wind speed affecting oil-immersed transformers, using experimental data and neural networks to model winding hotspot temperature, and studied the optimization of neural network structures and algorithms. Comparisons between measured and computed values using the recommended IEC60076 algorithm and the authors’ proposed neural network algorithm showed that values calculated using the neural network algorithm were closer to the measured values.

However, most of the aforementioned studies focused on model construction. The data they used mostly consisted of relevant data at the current moment of the transformer, with little attention paid to the hysteresis between the surface temperature of the transformer tank and the winding hotspot temperature (for example, the aforementioned studies by Deng et al. [10], Sun et al. [11], and Wei et al. [12]). Neglecting hysteresis may result in insufficient temperature feature information, thereby reducing the accuracy of the model inversion. Moreover, the interactions between different factors were ignored, and little research was conducted on the influence of different feature quantities on model performance (for example, the aforementioned studies by Sun et al. [11] and Wei et al. [12]). Inputting all monitoring data into the prediction model may increase complexity and affect performance. To overcome this issue, a correlation analysis should be conducted before inputting data into the model, selecting parameters most favorable for the prediction target to reduce redundancy and improve accuracy.

To address the aforementioned issues, this paper proposes a method for the inverse estimation of winding hotspot temperature in oil-immersed power transformers. Firstly, temperature rise experiments are conducted to collect raw data for inversion research, including temperatures at the inlet and outlet of the radiator, ambient temperature, top oil temperature, winding hotspot temperature, and surface temperature of the transformer tank. To tackle the issue of insufficient temperature feature information, the surface temperature data of the transformer tank are collected in the form of temperature images, fully considering the hysteresis of the data. Secondly, the mutual information algorithm is employed to analyze the correlation between each input feature quantity and the output feature quantity, obtaining the optimal feature set to address the problem of feature redundancy. Finally, based on the Gated Recurrent Unit (GRU), an inversion network is constructed. To overcome the relative weakness of the GRU in handling long-distance dependency relationships, Self-Attention (SA) is introduced for optimization, constructing a GRU-SA network for the inverse estimation of transformer winding hotspot temperature, achieving a high-precision estimation of transformer winding hotspot temperatures.

## 2. Analysis of Influencing Factors on Winding Hotspot Temperature

The inversion of winding hotspot temperature in transformers essentially involves establishing a functional relationship between its input feature vector and output feature vector. The influencing factors of winding hotspot temperature directly determine the accuracy of the model constructed, making the selection of influencing factors crucial.

A thermal circuit model for power transformers has been proposed in the literature [13], as illustrated in Figure 1.

In Figure 1, *P_1wdn_*, *P_2wdn_*, and *P_3wdn_* represent the winding losses of phases A, B, and C, respectively; *P_mp_*, *P_fe_*, and *P_tank_* represent the losses of the clamps, iron core, and tank, respectively; *C_1wdn_*, *C_2wdn_*, and *C_3wdn_* represent the thermal capacity of the phase A, B, and C windings, respectively; *C_mp_*, *C_fe_*, and *C_tank_* represent the thermal capacity of the clamps, iron core, and tank, respectively; *R_1wdn-oil_*, *R_2wdn-oil_*, and *R_3wdn-oil_* represent the thermal resistance of the phase A, B, and C windings to oil, respectively; *R_mp-oil_*, *R_fe-oil_*, and *R_tank-oil_* represent the thermal resistance of the clamps, iron core, and tank to oil, respectively; *R_tank-oil_* and *R_oil-radiator-air_* represent the thermal resistance of the tank and oil cooler to air, respectively; *θ_1wdn-hs_*, *θ_2wdn-hs_*, and *θ_3wdn-hs_* represent the winding hotspot temperatures of phases A, B, and C, respectively; *θ_top-oil_*, *θ_mp_*, and *θ_fe,_ θ_tank_* represent the top oil temperature, and the temperatures of the clamps, iron core, and tank, respectively.

The thermal resistances of the A, B, and C three-phase windings to oil, *R_1wdn-oil_*, *R_2wdn-oil_*, and *R_3wdn-oil_*, are represented by a thermal resistance *R_wdn-oil_*. The heat capacities *C_1wdn_*, *C_2wdn_*, and *C_3wdn_* of the A, B, and C three-phase windings are represented by the heat capacity *C_wdn_*. The winding losses *P_1wdn_*, *P_2wdn_*, and *P_3wdn_* of the A, B, and C three-phase windings are represented by a winding loss *P_wdn_*. Then, the three-phase branches on the left side of the thermal circuit model can be merged into one branch, resulting in a simplified thermal circuit model, as shown in Figure 2.

From Figure 2, we can derive two differential equations as follows in (1) and (2):(1)$$P_{wdn}=C_{wdn}\frac{d\theta_{wdn-hs}}{dt}+\frac{\theta_{wdn-hs}-\theta_{oil-top}}{R_{wdn-oil}}$$
(2)$$P_{tank}=C_{wdn}\frac{d\theta_{tank}}{dt}+\frac{\theta_{tank}-\theta_{air}}{R_{tank-air}}+\frac{\theta_{tank}-\theta_{oil-top}}{R_{tank-oil}}$$

Utilizing Euler’s formula, Equations (1) and (2) are discretized as shown in Equations (3) and (4):(3)$$\frac{d\theta_{wdn-hs}}{dt}\approx \frac{\theta_{wdn-hs,k}-\theta_{wdn-hs,k-1}}{T}$$
(4)$$\frac{d\theta_{tank}}{dt}\approx \frac{\theta_{tank,k}-\theta_{tank,k-1}}{T}$$

In Equations (3) and (4), *T* represents the sampling interval, i.e., the sampling period, and *k* denotes the index of discrete data. Thus, the differential equations can be discretized as shown in Equations (5) and (6):(5)$$P_{wdn}=C_{wdn}\frac{\theta_{wdn-hs,k}-\theta_{wdn-hs,k-1}}{T}+\frac{\theta_{wdn-hs}-\theta_{oil-top}}{R_{wdn-oil}}$$
(6)$$\begin{array}{l} P_{tank}=C_{wdn}\frac{\theta_{tank,k}-\theta_{tank,k-1}}{T}+\frac{\theta_{tank,k}-\theta_{air,k}}{R_{tank-air,k}}\\ \;\;+\frac{\theta_{tank,k}-\theta_{oil-top,k}}{R_{tank-oil,k}} \end{array}$$

Upon the simplification of Equations (5) and (6), we obtain Equations (7) and (8):(7)$$\begin{array}{ll} \theta_{wdn-hs,k} & =\frac{C_{wdn}R_{wdn-oil,k}}{C_{wdn}R_{wdn-oil,k}+T}\theta_{wdn-hs,k-1}\\ & +\frac{T}{C_{wdn}R_{wdn-oil,k}+T}\theta_{oil-top,k}\\ & +\frac{R_{wdn-oil,k}T}{C_{wdn}R_{wdn-oil,k}+T}P_{wdn} \end{array}$$
(8)$$\begin{array}{ll} \theta_{oil-top,k} & =\frac{C_{wdn}R_{tank-air,k}R_{tank-oil,k}+TR_{tank-air,k}+TR_{tank-oil,k}}{TR_{tank-air,k}}\theta_{tank,k}\\ & -\frac{C_{wdn}R_{tank-air,k}R_{tank-oil,k}}{TR_{tank-air,k}}\theta_{tank,k-1}-\frac{TR_{tank-oil,k}}{TR_{tank-air,k}}\theta_{air,k}\\ & -\frac{TR_{tank-air,k}R_{tank-oil,k}}{TR_{tank-air,k}}P_{tank} \end{array}$$

Setting $K_{1}=\frac{C_{wdn}R_{wdn-oil,k}}{C_{wdn}R_{wdn-oil,k}+T}$, $K_{2}=\frac{T}{C_{wdn}R_{wdn-oil,k}+T}$, $K_{3}=\frac{R_{wdn-oil,k}T}{C_{wdn}R_{wdn-oil,k}+T}$, $K_{4}=\frac{C_{wdn}R_{tank-air,k}R_{tank-oil,k}+TR_{tank-air,k}+TR_{tank-oil,k}}{TR_{tank-air,k}}$, $K_{5}=-\frac{C_{wdn}R_{tank-air,k}R_{tank-oil,k}}{TR_{tank-air,k}}$, $K_{6}=-\frac{TR_{tank-oil,k}}{TR_{tank-air,k}}$, $K_{7}=-\frac{TR_{tank-air,k}R_{tank-oil,k}}{TR_{tank-air,k}}$, Equations (7) and (8) can be expressed as Equations (9) and (10):(9)$$\theta_{wdn-hs,k}=K_{1}\theta_{wdn-hs,k-1}+K_{2}\theta_{oil-top,k}+K_{3}P_{wdn}$$
(10)$$\theta_{oil-top,k}=K_{4}\theta_{tank,k}+K_{5}\theta_{tank,k-1}+K_{6}\theta_{air,k}+K_{7}P_{tank}$$
where *K*_1_*~K*_7_ can be estimated using regression algorithms. Substituting Equation (10) into Equation (9) yields Equation (11):(11)$$\begin{array}{ll} \theta_{wdn-hs,k} & =K_{1}\theta_{wdn-hs,k-1}+K_{2}K_{4}\theta_{tank,k}+K_{2}K_{5}\theta_{tank,k-1}\\ & +K_{2}K_{6}\theta_{air,k}+K_{2}K_{7}P_{tank}+K_{3}P_{wdn} \end{array}$$

From Equations (9) and (11), it can be observed that the transformer winding hotspot temperature is related to the top oil temperature, ambient temperature, tank temperature, and the temperature of the tank at the previous time step. Equation (11) also demonstrates the hysteresis effect of the tank temperature on the winding hotspot temperature.

To reduce the top oil temperature, oil is typically circulated into coolers for cooling. It is evident that the cooler is the most crucial device for dissipating internal heat in the transformer. Therefore, the variations in temperatures at the outlet and inlet of the cooler are closely related to the changes in the transformer winding hotspot temperature.

Hence, this paper considers the top oil temperature, ambient temperature, temperatures at the outlet and inlet of the cooler, tank temperature, and the temperature of the tank at the previous time step as feature parameters for the inversion of transformer winding hotspot temperature.

## 3. Inversion Method

In this section, based on the advantages of the mutual information (MI), Self-Attention (SA), and Gated Recurrent Unit (GRU) algorithms [14,15,16], we propose a transformer winding hotspot temperature inversion method. By leveraging the MI, SA, and GRU algorithms, we aim to enrich the information content of the inversion input features while eliminating feature redundancy, thereby ensuring good inversion results.

The specific data collection, training, and inversion process are illustrated in Figure 3.

Step 1: Design temperature rise experiments and collect relevant temperature raw data using fiber optic sensors, thermocouples, and infrared thermal imagers.

Step 2: Clean the raw data set by handling abnormal outliers and missing values. Extract temperature information from infrared images to obtain the top oil temperature, ambient temperature, the maximum, minimum, and average temperatures of the tank side and top as input features for inversion, and the winding hotspot temperature as the output feature for inversion.

Step 3: Utilize the SA module to optimize the GRU network and construct the SA-GRU network for inversion.

Step 4: Employ the MI method to select input feature parameters, reducing the dimensionality of input features and obtaining the optimal feature subset for inversion.

Step 5: Divide the dimensionality-reduced dataset into two sequence sets, serving as the training set and test set, respectively, and input them into the SA-GRU network for training and testing.

Step 6: Analyze the inversion results and compare them with the measured results to evaluate the model performance.

### 3.1. Data Collection for Temperature Rise Experiment

In this study, temperature rise tests were conducted on an SFZ-40000/110 oil-immersed 110 kV transformer on-site. The main parameters are shown in Table 1.

Through the placement of fiber optic sensors at internal relevant positions, the winding hotspot temperature and top oil temperature are measured, as depicted in Figure 4. Fiber optic probes for measuring the top oil temperature are placed on the wire clamp at the fixed neutral point. Since the hotspot temperature of the high-voltage winding of the transformer is generally higher than that of the low-voltage winding, this study considered the hotspot temperature of the low-voltage winding as the hotspot temperature of the entire winding. According to the literature, the position of the transformer winding hotspot temperature is approximately at 90% of the winding height. Therefore, fiber optic probes are placed at 90% of the low-voltage winding to measure the winding hotspot temperature [17,18].

The ambient temperature is obtained by placing thermocouples at the four corners of the transformer, and the temperatures at the inlet and outlet of the cooler are also measured by placing thermocouples near the inlet and outlet.

To increase the information content of the tank temperature, infrared thermal imagers were used to obtain temperature images of the top and side of the tank, which contained temperature information for the entire surface. The temperature information in the thermal images was more abundant, facilitating better analysis of the potential correlation between tank temperature and winding hotspot temperature. Two infrared thermal imagers were used to collect temperature images of the top and side of the tank, as depicted in the layout diagram in Figure 5.

According to the specifications of the temperature rise test in IEC 60076-2 [19], three load levels of 50%, 75%, and 100% were set. Experimental data were collected every 5 min.

### 3.2. Data Preprocessing

The experimental raw data were subjected to data cleaning to remove outliers and missing values. To standardize the format of input feature quantities, temperature image data were converted into temperature sequence data.

Initially, the temperature images and the coordinates of the positions of the tank side and top were inputted into the Segment Anything Model (SAM) to segment the tank side and top parts [20]. Subsequently, the temperature matrix corresponding to the resulting temperature image was obtained, providing the temperature values of all points on the tank side and top. Finally, the maximum, minimum, and average temperature values of each point on the side and top were extracted as their temperature features, thereby obtaining the maximum, minimum, and average temperature values of the tank side and top at each sampling time.

Since four thermocouples were used to measure the ambient temperature during the experiment, there were four ambient temperature values at each sampling time. The average value of these four values was taken as the ambient temperature at the current time.

In addition to the experimental measurements of the temperatures at the inlet and outlet of the cooler and the top oil temperature, a total of 16 feature quantities were obtained, including the maximum, minimum, and average temperature values of the tank side and top, the maximum, minimum, and average temperature values of the tank side and top at the previous time step, ambient temperature, temperatures at the inlet and outlet of the cooler, and top oil temperature, as candidate input feature quantities. The winding hotspot temperature was considered as the output feature quantity. For ease of representation, each feature was given a name, as shown in Table 2, and some sample data are presented in Figure 6. The entire data processing process took a week.

### 3.3. Feature Selection

Using all feature parameters for inversion may lead to feature redundancy and increase the complexity of the inversion model. Before inversion, conducting a correlation analysis between the input and output feature quantities and selecting feature quantities with a high correlation with the output feature can effectively improve the performance of the inversion model.

The mutual information (MI) algorithm is a commonly used feature selection algorithm used to evaluate the correlation between features and the target variable. It measures the degree of dependence between them by calculating the mutual information between the feature and the target variable, thereby determining the most relevant features.

The basic idea of the MI algorithm is to measure the difference between the joint probability distribution of the feature and the target variable and their respective marginal probability distributions to quantify their correlation. If there is high mutual information between the feature and the target variable, it indicates a high degree of dependence between them, and the feature is useful for predicting the target variable.

The mutual information between two discrete random variables *X* and *Y* can be defined as follows [21]:(12)$$I(X;Y)=\sum_{y\in Y}\sum_{x\in X}p(x,y)\log(\frac{p(x,y)}{p(x)p(y)})$$

In Equation (12), *p*(*x,y*) represents the joint probability distribution of *X* and *Y*, while *p*(*x*) and *p*(*y*) are the marginal probability distribution functions of *X* and *Y*, respectively.

In this study, the MI algorithm was utilized to select features among sixteen input features, obtaining the optimal feature subset that maximizes the accuracy of inversion. The feature selection process is illustrated in Figure 7.

The specific steps are as follows: (1) data preprocessing: determine sixteen parameters including environmental temperature, the temperatures of the radiator outlet and inlet, the maximum temperature of the oil tank side surface, etc., as candidate input feature parameters; (2) calculate the mutual information (MI) values between the candidate input feature parameters and winding hotspot temperature separately; (3) arrange candidate input feature parameters in descending order of MI values; (4) select 1–16 input features according to the order and input them into the inversion network for training, recording the inversion errors corresponding to the different numbers of input features; (5) select the combination of input features corresponding to the minimum inversion error as the optimal feature set for the inversion target.

### 3.4. GRU-SA Inversion Network

The GRU (Gated Recurrent Unit) [22] is a variant of recurrent neural networks (RNNs) that can be utilized for processing sequential data such as text, speech, and time series. It was proposed by Cho et al. in 2014 [22]. Compared to traditional RNN models, GRU exhibits stronger modeling capabilities and better gradient propagation properties. The GRU network enables bidirectional information propagation between layers, allowing for the persistence of information, thus endowing the network with long-term memory capabilities. Figure 8 illustrates the specific structure of the GRU [23].

From the figure, it can be observed that the GRU is equipped with two gates: the reset gate *r_t_* and the update gate *z_t_*, both of which are used for information filtering. Firstly, the GRU will calculate the reset gate and update gate based on the current input and the previous hidden state. Then, the GRU will update the memory unit based on the gate control signal. The reset gate *r_t_* controls how many previous hidden states should be forgotten, and if the reset gate is close to 0, it ignores the previous information and it only focuses on the current input. If the value of the reset gate is close to 1, the previous information is retained and combined with the current input information. While the update gate *z_t_* determines how much new information should be retained and is combined with the input signal to generate the candidate hidden state, if the value of the update gate is close to 0, then the previous information is retained and used as the new memory unit. If the value of the update gate is close to 1, it means that the previous information is replaced with the current input information as the new memory unit. The update gate *z_t_* then filters and determines how much of the candidate hidden state *ĥ_t_* and the previous state *h_t__−1_* should be preserved, thereby deciding the amount of past state information retained in the current state *h_t_*. Finally, the GRU outputs the hidden state of the current moment based on the updated memory unit. The entire process is formulated as follows [24]:

(1) Update Gate:(13)$$r_{t}=f(x_{t}\omega_{xr}+h_{t-1}\omega_{hr}+b_{r})$$

(2) Reset Gate:(14)$$z_{t}=f(x_{t}\omega_{xz}+h_{t-1}\omega_{hz}+b_{z})$$

(3) Filtering and Memory of Previous Information and Current Input to Obtain Hidden State:(15)$${\hat{h}}_{t}=\tanh(x_{t}\omega_{xh}+(r_{t}⊙h_{t-1})\omega_{hh}+b_{h})$$

(4) Update Memory to Obtain Current State:(16)$$h_{t}=(1-z_{t})⊙h_{t-1}+z_{t}⊙{\hat{h}}_{t}$$

Whereas *w* and *b* represent the weights and biases of the network, *f*(*·*) and tanh denote the activation functions, and ⊙ refers to the Hadamard product.

Although GRU can handle sequential data and capture dependencies, it is relatively weaker in understanding long-range dependencies due to the lack of a Self-Attention mechanism. Additionally, the model’s expressive capacity of GRU is relatively limited, leading to deficiencies in handling complex tasks. To address these issues, this paper introduces SA to optimize the GRU.

SA weighs each element in the input sequence and uses the weighted results as feature representations to capture the correlations between different positions in the sequence. The computational formula for the Self-Attention mechanism is as follows [25,26,27]:(17)$$\mathrm{Attention}(Q,K,V)=\mathrm{softmax}(\frac{QK^{T}}{\sqrt{d_{k}}})V$$
in which, *Q*, *K*, and *V* represent the query vector, key vector, and value vector, respectively; softmax indicates the normalization of the attention weights; and *d_k_* represents the dimension of the vectors.

The network structure after introducing SA is shown in Figure 9, and this network is referred to as the GRU-SA network.

The input sequence undergoes feature extraction via multiple GRU layers, followed by global contextual modeling through the SA layer. Finally, a linear layer is applied for transformation. The input dimension of the linear layer is the same as the hidden layer dimension in the GRU module, with an output dimension of 1. This effectively captures the temporal information, dependency, and global semantics in the sequence, yielding the final output.

### 3.5. Evaluation Metrics for Inversion Model

In order to reflect the dispersion of errors, the mean squared error (MSE) is utilized. The mean absolute error (MAE) is employed to measure the average error. The coefficient of determination (R^2^) and the mean absolute percentage error (MAPE) are used to assess the fitting degree of the model. To comprehensively evaluate the performance of the model, this study employs these four metrics: MSE, MAE, and MAPE should be minimized, while R^2^ should be maximized for better model performance. The computational formulas are represented in Equations (18)–(21) [28,29]:(18)$$\mathrm{MSE}=\frac{\sum_{i=1}^{n}{\left(x(i)-\hat{x}(i)\right)}^{2}}{n}$$
(19)$$\mathrm{MAE}=\frac{\sum_{i=1}^{n}\left|x(i)-\hat{x}(i)\right|}{n}$$
(20)$$R^{2}=1-\frac{\sum_{i=1}^{n}{\left(x(i)-\hat{x}(i)\right)}^{2}}{\sum_{i=1}^{n}{\left(x(i)-\bar{x}(i)\right)}^{2}}$$
(21)$$\mathrm{MAPE}=\frac{100\%}{n}\sum_{i=1}^{n}\left|\frac{x(i)-\hat{x}(i)}{x(i)}\right|$$

Here, *n* denotes the number of samples, and x(i), $\hat{x}(i)$, and $\bar{x}(i)$ represent the actual value, predicted value, and mean value of the *i* sample, respectively.

## 4. Experimental Validation

Collect a set of data every 5 min according to the method in Section 3.1. After data preprocessing using the method mentioned in Section 3.2, a total of 471 sets of data were obtained. Among the 10 sets of data collected every 50 min, take the first set as the test set, that is, divide all data into a training set and a testing set at a ratio of 9:1. Within the training set, 20% is randomly selected as the validation set. To enhance the training efficiency of the model, all data are normalized.

The experimental setup included an Intel Core i5-7400 CPU and Windows 10 operating system. The program was written in Python, utilizing the PyTorch (https://pytorch.org/ accessed on 17 July 2024) deep learning framework, with Python version 3.8.

### 4.1. Consideration of Lagged Data for Inversion Effectiveness

To demonstrate the inversion of the transformer winding hotspot temperature considering the lagged effect of the oil tank temperature, two sets of input feature parameters are employed for inversion experiments in this section: one set does not consider the temperature of the tank at the previous time step, comprising ten input features; the other set considers the temperature of the tank at the previous time step, comprising a total of sixteen input features. Table 3 presents the error analysis results for the two sets of experiments.

From Table 3, it can be observed that when considering the lagged effect of the oil tank surface temperature, the inversion curve becomes closer to the actual curve, and the error metrics also improve. Specifically, MSE, MAE, R^2^, and MAPE increase by 47.83%, 0.76%, 1.84%, and 3.44%, respectively, indicating better model performance. This improvement is attributed to the enrichment of information features in the input features after considering the lagged temperature data, which establishes potential associative relationships between these information features and the inversion target parameters. By incorporating this information into the model’s input features, the model can better utilize such associative relationships and learn more accurate and effective feature representations. Consequently, the model can better understand and capture both local and global features in the input data, thereby accomplishing the inversion task more effectively.

Thus, incorporating the lagged effect of the oil tank surface temperature into the input features significantly enhanced the performance of the inversion model for predicting transformer winding hotspot temperatures, providing more accurate and reliable inversion results.

### 4.2. Feature Dimensionality Reduction

To determine which input features have greater correlation with the winding hotspot temperature, the mutual information (MI) method is employed to calculate the correlation between the input features and the output feature. The MI values of the 16 input features with the winding hotspot temperature are illustrated in Figure 10.

Each feature contributes differently to the improvement of model performance. In order to reduce the redundancy of input features and the complexity of model training while ensuring model accuracy, the GRU-SA network is employed as the inversion network according to the order calculated by the MI. The inversion errors of models with different numbers of input features are computed. Figure 11 illustrates the relationship between the inversion errors and the number of input features.

From Figure 9, it can be observed that for the inversion task of winding hotspot temperature, selecting the first 13 input features yields optimal results. Therefore, the optimal feature set for winding hotspot temperature is as follows:$$\left\{\begin{array}{l} \mathrm{RO},\mathrm{TOT},\mathrm{RI},T\text{-}\mathrm{AVE},T\text{-}\mathrm{MAX},T\text{-}\mathrm{AVE}\text{-}P,T\text{-}\mathrm{MAX}\text{-}P,\\ \mathrm{AT},B\text{-}\mathrm{MAX},T\text{-}\mathrm{MIN},T\text{-}\mathrm{MIN}\text{-}P,B\text{-}\mathrm{AVE},B\text{-}\mathrm{MAX}\text{-}P \end{array}\right\}$$

To validate the effectiveness and superiority of the MI method for feature dimensionality reduction, a comparison is made between the MI method, the no-feature selection method, and the Spearman method [30]. The inversion error results for different optimization methods are presented in Table 4.

From Table 4, it can be observed that compared to the method without feature selection, the RMSE, MAE, R^2^, and MAPE metrics are improved by 12.5%, 4.9%, 0.25%, and 6.01%, respectively, when using the MI method. Compared to the Spearman method, the respective metrics are improved by 11.27%, 3.4%, 0.22%, and 3.81%.

The MI method considers the non-linear relationships between features when selecting features, which enables it to effectively capture the complex associations between input features and target variables. Therefore, after using the MI method for feature selection, the model can conduct inversion more accurately, resulting in higher inversion accuracy.

The inversion results using the optimal feature set are illustrated in the following figure. From Figure 12, it can be observed that the trend of the inversion value is very close to the actual value. Hence, the MI method can eliminate redundancy in input features, further enhancing model performance.

### 4.3. Parameter Optimization of the Model

Through multiple experiments, it was found that the parameters hidden size and num layers in the model had a significant impact on the model’s performance. Hidden size defines the size of the hidden layer in the model, which determines the model’s representational and learning abilities. A larger hidden size can increase the network’s expressive power but also increases the model’s complexity and computational cost. Num layers defines the number of layers in the GRU unit of the model. Increasing the number of layers can enhance the model’s expressive power, enabling it to better capture and represent complex sequence patterns and features, but it also increases the model’s complexity and training time.

To find the optimal parameter combination, the particle swarm optimization algorithm [31] was employed to optimize the parameters of the GRU-SA model, with the minimum MSE value as the optimization target. The range was set to [32, 128] for hidden size and [1, 4] for num layers. The optimization results are shown in Figure 13.

From the above results, it can be observed that when setting 1 to 64 and 2 to 2, the model performance reaches its optimum.

To further demonstrate the superiority of the proposed method in this paper, comparative experiments were conducted with existing typical methods, namely the RNN, LSTM, and the original GRU. To ensure the fairness of model comparison, the optimal feature set obtained by the MI algorithm was used as the input feature set, and the particle swarm algorithm was used to optimize the parameters of each model. The inversion results are shown in Figure 14, and the inversion errors of each model are shown in Table 5.

It can be observed that compared with other typical methods, the inversion results of the proposed method in this paper exhibit the smallest deviation from the actual values, enabling better inversion of the target parameters. From Figure 14, it can be seen that compared with other typical methods, the inversion value of this method and the actual value had the closest trend, indicating that the proposed method can more accurately invert the trend of winding hot spot temperature changes. Compared with the RNN method, the proposed method in this paper showed improvements of 79.32%, 35.51%, 5.12%, and 32.27% in terms of the MSE, MAE, R^2^, and MAPE, respectively. Compared with the LSTM and GRU, the proposed method showed improvements of 84.2%, 41.95%, 7.31%, 3.95%, and 61.04%, 24.37%, 2.05%, and 19.38% for each indicator, respectively.

In order to compare the training and evaluation time of different models, train and evaluate each model optimized by the parameters using the optimal feature set. The time required for each model is shown in Table 6.

According to Table 6, the training and evaluation time for each model was within 10 s, and the speed was relatively fast. Relatively speaking, the GRU-SA proposed in this article took a shorter amount of time and had higher accuracy than the LSTM and GRU. Although the GRU-SA took slightly longer than the RNN, it could greatly improve inversion accuracy with a slight sacrifice of speed.

Compared with traditional models, when using the proposed method in this paper for sequence modeling tasks, the model demonstrated stronger understanding and modeling capabilities of input sequences, better handling of long-term dependencies, nonlinear relationships, and information loss issues, resulting in higher performance in sequence modeling tasks. Therefore, the transformer winding hotspot temperature inversion method proposed in this paper based on the GRU-SA can achieve better inversion results while reducing model redundancy and complexity.

In order to compare and analyze the trend of changes in the actual and inverted values of hot spot temperatures, this study calculated the temperature difference at each time interval of the two temperature values, as shown in Figure 15.

From Figure 15, it can be seen that the trend of actual values and inversion values was basically consistent, indicating that the method proposed in this paper can accurately invert the trend and temperature values of winding hot spot temperature changes.

## 5. Conclusions

This paper proposed a method for transformer winding hotspot temperature inversion, from which the following four conclusions are drawn:(1)Considering the lagged data of the oil tank temperature could enrich the information content of the data and more accurately invert the transformer winding hotspot temperature.(2)Using the MI algorithm for feature selection and dimensionality reduction of input features could reduce model redundancy and complexity, thereby improving the inversion accuracy of the model.(3)The GRU-SA inversion network proposed in this paper, by introducing SA into the GRU, enabled the network to better capture the correlation between different positions in the sequence data, and compared with traditional networks, the GRU-SA exhibited better performance in inversion tasks.(4)The effectiveness of the proposed method was validated through field temperature rise experimental data of a 110 kV transformer.

In future research, more environmental factors can be considered to study the hot spot temperature inversion method for transformers with more complex operating environments.

## Figures and Tables

**Figure 1 sensors-24-04734-f001:**
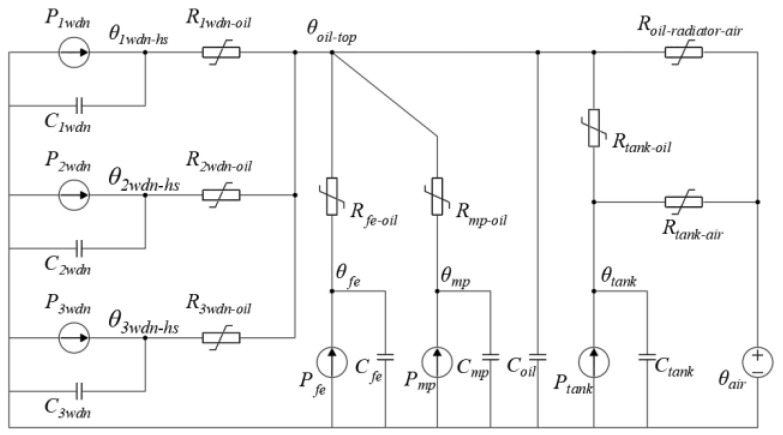
Thermal circuit model of transformer.

**Figure 2 sensors-24-04734-f002:**
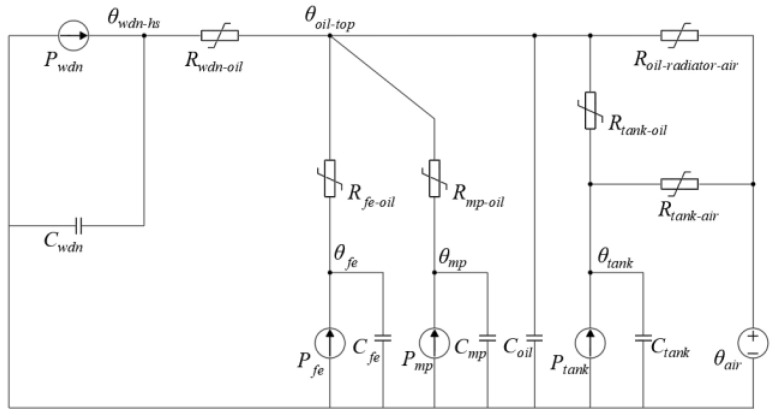
Simplified thermal circuit model.

**Figure 3 sensors-24-04734-f003:**
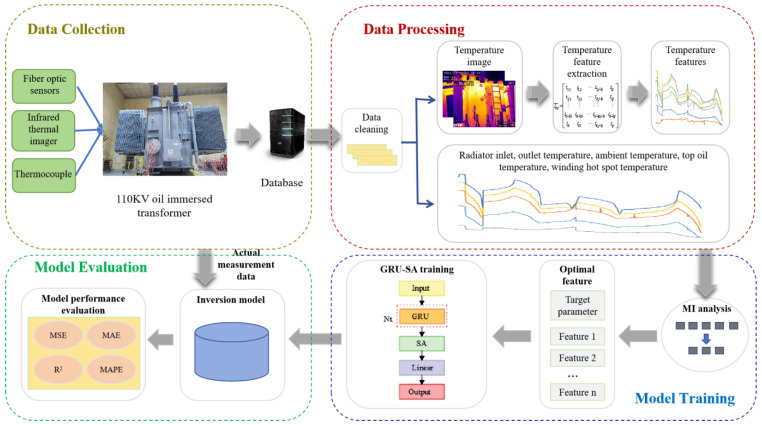
Flowchart of the inversion method.

**Figure 4 sensors-24-04734-f004:**
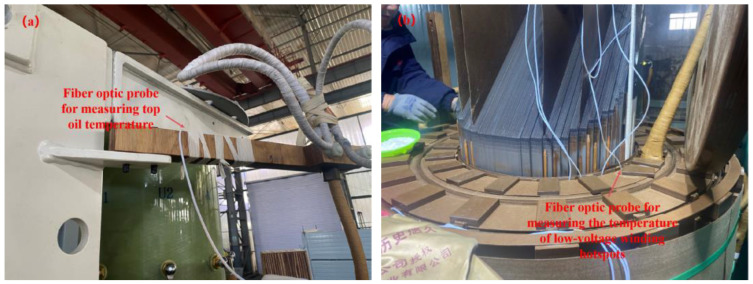
Layout of fiber optics: (**a**) fiber for measuring top oil temperature; (**b**) fiber for measuring winding hotspot temperature.

**Figure 5 sensors-24-04734-f005:**
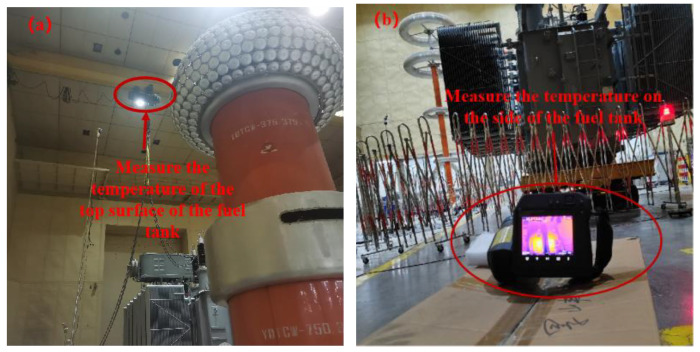
Layout diagram of infrared thermal imagers: (**a**) top of the tank; (**b**) side of the tank.

**Figure 6 sensors-24-04734-f006:**
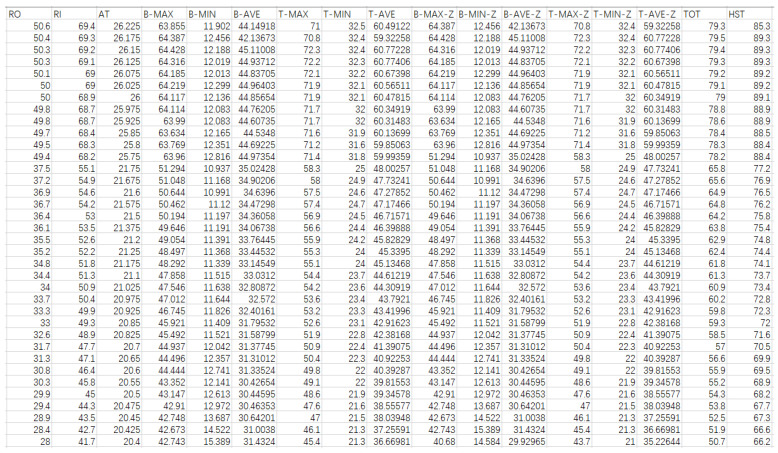
Sample of experimental data.

**Figure 7 sensors-24-04734-f007:**
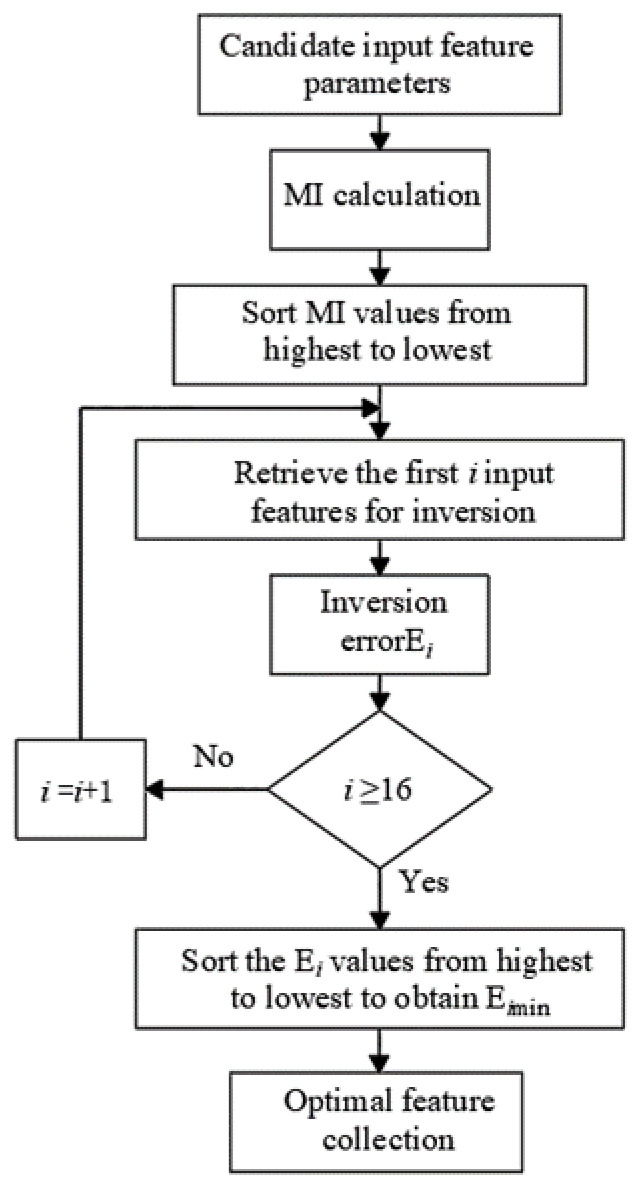
Flowchart of feature selection process.

**Figure 8 sensors-24-04734-f008:**
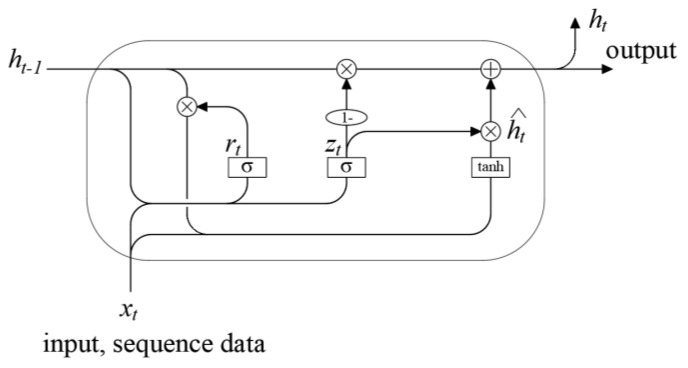
Structure of GRU.

**Figure 9 sensors-24-04734-f009:**
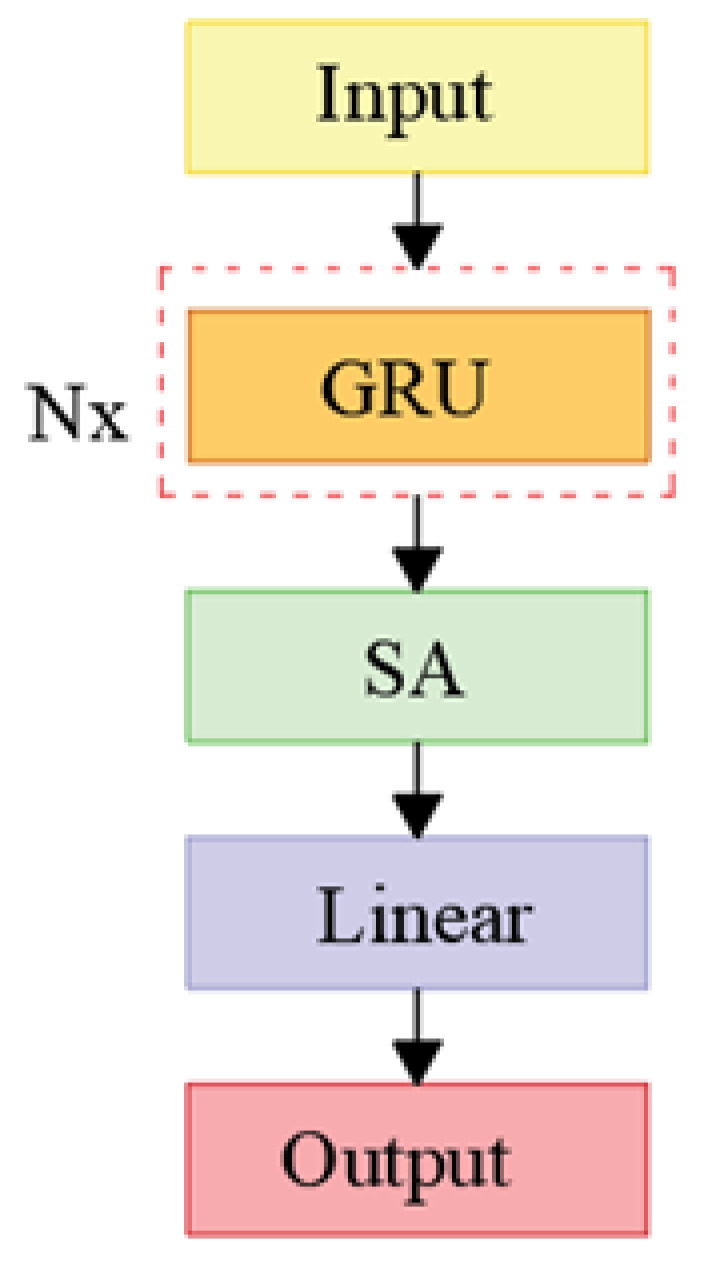
Illustrates the architecture of the GRU-SA network.

**Figure 10 sensors-24-04734-f010:**
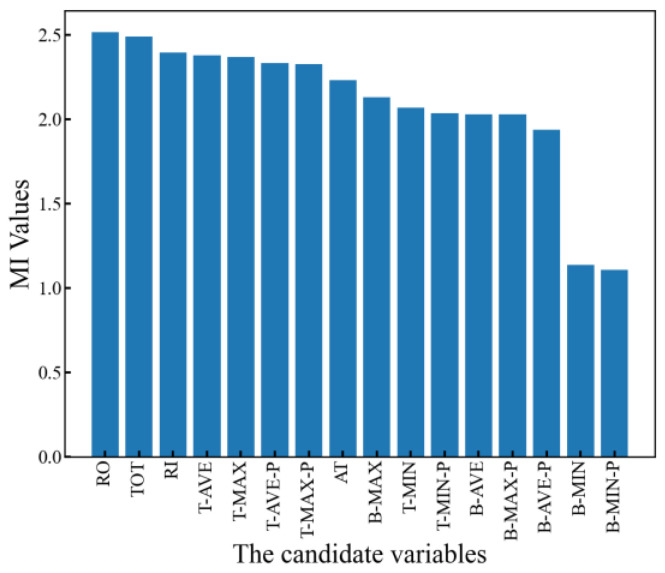
MI calculation results between input features and winding hotspot temperature.

**Figure 11 sensors-24-04734-f011:**
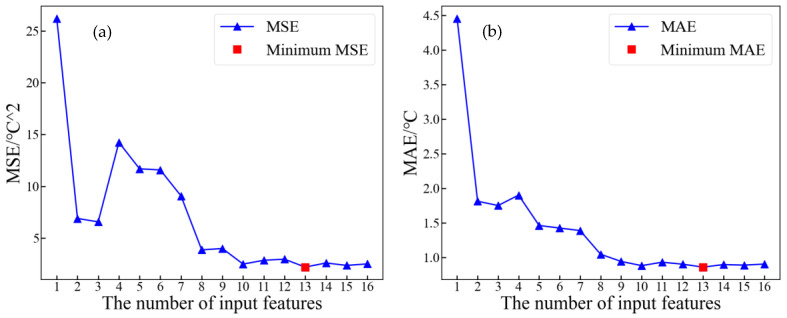
Relationship between model inversion errors and number of input features: (**a**) MSE; (**b**) MAE; (**c**) R^2^; (**d**) MAPE.

**Figure 12 sensors-24-04734-f012:**
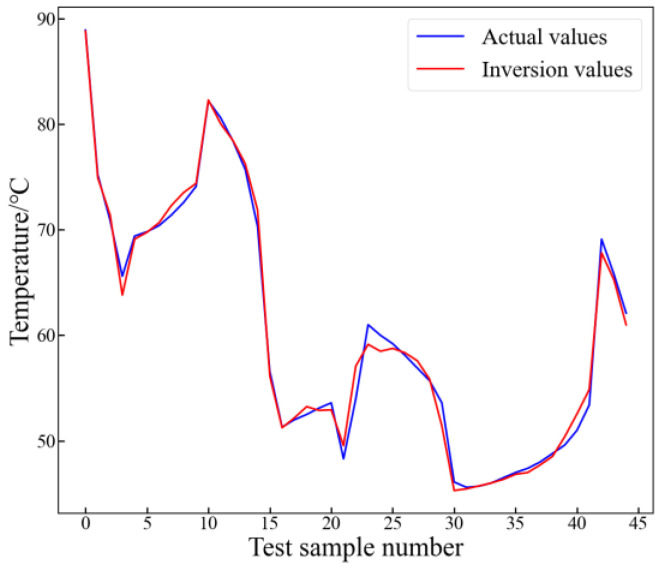
Inversion results with optimal feature set.

**Figure 13 sensors-24-04734-f013:**
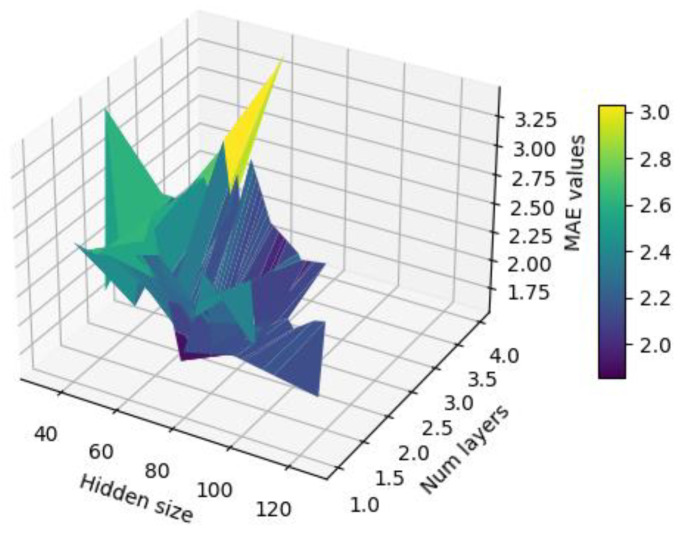
MAE values of the inversion model under different parameter settings.

**Figure 14 sensors-24-04734-f014:**
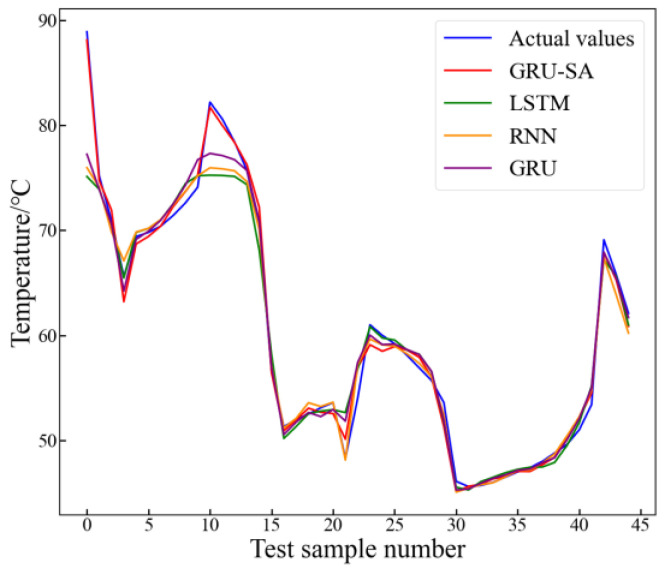
Comparison of results from different inversion methods.

**Figure 15 sensors-24-04734-f015:**
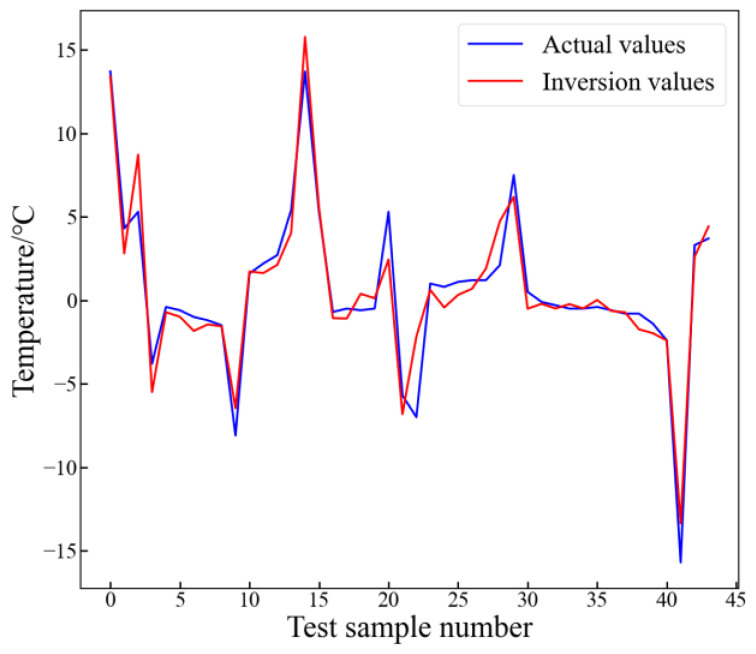
Trend of changes in actual and inversion values of hot spot temperatures.

**Table 1 sensors-24-04734-t001:** Main parameters.

Parameters	Value	Parameters	Value
Rated capacity	40,000 kVA	Rated voltage	110 kV ± 8 × 1.25%/10 kV
Rated current	209.95 A/2199.43 A	Connection group	YNd 11
Numberof.turns·in·high-voltage	635 turns	Numberofturns·in low-voltage	105 turns

**Table 2 sensors-24-04734-t002:** Feature naming.

Feature	Name	Feature	Name
Maximum temperature of the tank side	B-MAX	Maximum temperature of the tank top at the previous time step	T-MAX-P
Minimum temperature of the tank side	B-MIN	Minimum temperature of the tank top at the previous time step	T-MIN-P
Average temperature of the tank side	B-AVE	Average temperature of the tank top at the previous time step	T-AVE-P
Maximum temperature of the tank top	T-MAX	Ambient temperature	AT
Minimum temperature of the tank top	T-MIN	Temperature at the outlet of the cooler	RO
Average temperature of the tank top	T-AVE	Temperature at the inlet of the cooler	RI
Maximum temperature of the tank side at the previous time step	B-MAX-P	Top oil temperature	TOT
Minimum temperature of the tank side at the previous time step	B-MIN-P	Winding hotspot temperature	HST
Average temperature of the tank side at the previous time step	B-AVE-P		

**Table 3 sensors-24-04734-t003:** Model errors with different input features.

Output Feature	Input Features	MSE/°C	MAE/°C	R^2^	MAPE/%
HST	Without lagged effect	4.7943	0.9108	0.9631	1.6694
	With lagged effect	2.5010	0.9039	0.9808	1.6120

**Table 4 sensors-24-04734-t004:** Comparison of errors with different optimization methods.

	Input Features	MSE/°C	MAE/°C	R^2^	MAPE/%
HST	No Feature Selection	2.5010	0.9039	0.9808	1.6120
	Spearman	2.4663	0.8899	0.9810	1.5752
	MI	2.1883	0.8596	0.9832	1.5152

**Table 5 sensors-24-04734-t005:** Performance comparison of different inversion methods for hotspot temperature inversion.

	Inversion Method	MSE/°C	MAE/°C	R^2^	MAPE/%
HST	RNN	7.9371	1.2629	0.9389	2.0629
	LSTM	10.3869	1.4032	0.9201	2.3086
	GRU	4.2126	1.0769	0.9676	1.7331
	GRU-SA	1.6414	0.8145	0.9874	1.3973

**Table 6 sensors-24-04734-t006:** Training and evaluation time for each model.

Model	Time/s
RNN	3.3191
LSTM	7.3678
GRU	9.2891
GRU-SA	5.7304

## Data Availability

Data associated with this research are available and can be obtained by contacting the corresponding author upon reasonable request.

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
