# Peer review of "Inversion Method for Transformer Winding Hot Spot Temperature Based on Gated Recurrent Unit and Self-Attention and Temperature Lag"

_sensors, 2024, doi:10.3390/s24144734_

Round 1

Reviewer 1 Report

Comments and Suggestions for Authors

Congratulations on the article for presenting a very current topic in the area of ​​power systems.

 The hot spot temperature of transformer windings is an important indicator for measuring 11 insulation performance, and its accurate inversion is crucial to ensure timely and accurate fault pre-12 diction of transformers

Congratulations on the article for presenting excellent English.

Reviewer 2 Report

Comments and Suggestions for Authors

1. I recommend that the titles do not have acronyms because it is rare for readers to use that word to perform the search.

2. What damage is suffered by a transformer that operates for 6 months with a hot spot of 100 oC? “If the winding hotspot temperature is too high, it will affect the equipment's voltage with stand capability and mechanical strength, leading to breakdown accidents [4,5].”

3. The authors should add a paragraph mentioning the methods that exist to calculate transformer hot spots.

4. Correct the position of the point in “Deng et al [10]. “and review the entire manuscript.

5. The authors must specify which articles have the disadvantages they mention below in a table (for example). “However, most of the aforementioned studies focused on model construction. The data they used mostly consisted of relevant data at the current moment of the transformer, with little attention paid to the hysteresis between the surface temperature of the transformer tank and the winding hotspot temperature. Neglecting hysteresis may result in insufficient temperature feature information, thereby reducing the accuracy of the model inversion. Furthermore, the interactions between different factors were ignored, and little research was conducted on the influence of different feature quantities on model performance. Inputting all monitoring data into the prediction model may increase complexity and affect performance.”

6. Authors have to explain how this operation was carried out so that readers understand. “and merging the three-phase branches into one branch, the simplified thermal circuit model is obtained as shown in Figure 2.”

7. The authors have to give more details of the linear layer. “Finally, a linear layer is applied for transformation, effectively capturing temporal information, dependencies, and global semantics within the sequence, thus yielding the ultimate output”

8. What type of transformers does this apply to? Make a drawing if the transformer is single-phase or three-phase. “According to the literature, the position of the transformer winding hotspot temperature is approximately at 90% of the winding height.”

9. Authors should use space between quantity and unit and review the entire document “635turns”

10. What does it mean that equation (15) uses the Hadamard product?

11. Why is i greater than or equal to 16 in Figure 7 (Flowchart of feature selection process? What are the input data of figure 7?

12. Include future work of this research in the conclusions.

Comments on the Quality of English Language

Minor editing of English language required

Reviewer 3 Report

Comments and Suggestions for Authors

Round 2

Reviewer 3 Report

Comments and Suggestions for Authors
